# Effects of Dietary Amylose—Amylopectin Ratio on Growth Performance and Intestinal Digestive and Absorptive Function in Weaned Piglet Response to Lipopolysaccharide

**DOI:** 10.3390/ani12141833

**Published:** 2022-07-19

**Authors:** Min Wang, Can Yang, Qiye Wang, Jianzhong Li, Yali Li, Xueqin Ding, Pengfei Huang, Huansheng Yang, Yulong Yin

**Affiliations:** 1Hunan International Joint Laboratory of Animal Intestinal Ecology and Health, College of Life Sciences, Hu-nan Normal University, Changsha 410081, China; wangmin202205@163.com (M.W.); yangcansky@163.com (C.Y.); wangqiye@hunnu.edu.cn (Q.W.); ljzhong@hunnu.edu.cn (J.L.); yalili@hunnu.edu.cn (Y.L.); xueqinding@hunnu.edu.cn (X.D.); perfehuang@foxmail.com (P.H.); yinyulong@isa.ac.cn (Y.Y.); 2Laboratory of Animal Nutrition and Human Health, College of Life Sciences, Hunan Normal University, Changsha 410081, China; 3College of Life Sciences, Hunan Provincial Key Laboratory of Biological Resources, Protection and Utilization in Nanyue Mountain Area, Hengyang Normal University, Hengyang 421008, China; 4Hunan Provincial Key Laboratory of Animal Nutritional Physiology and Metabolic Process, Scientific Observing and Experimental Station of Animal Nutrition and Feed Science in South-Central, Minis-try of Agriculture, Changsha 410125, China; 5Hunan Provincial Engineering Research Center for Healthy Livestock and Poultry Production, Key Laboratory of Agro-ecological Processes in Subtropical Region, Institute of Subtropical Agriculture, Chinese Academy of Sciences, Changsha 410125, China; 6State Key Laboratory of Applied Microbiology Southern China, Guang-dong Institute of Microbiology, Guangdong Academy of Sciences, Guangzhou 510070, China; 7Guangdong Provincial Key Laboratory of Microbial Culture Collection and Application, Guangdong Institute of Microbiology, Guangdong Academy of Sciences, Guangzhou 510070, China

**Keywords:** amylose–amylopectin, LPS, growth performance, intestinal digestion and absorption, weaned piglets

## Abstract

**Simple Summary:**

As one of the most important components, starch has made great contributions to feed but was wasted numerously during pig feeding. This paper is designed to explore the optimal ratio of dietary amylose to amylopectin for gut health and absorption, thereby improving starch utilization. Our results indicated that the dietary amylose/amylopectin ratio (AAR) of 0.60 could reduce feed conversion rate of piglets, having a certain positive significance for saving feed. In addition, under lipopolysaccharide (LPS) stress, a diet with an AAR of 0.40 to 0.60 significantly improved the intestinal health of piglets, which would provide data to support for the formulation of feed in weaned piglets during bacterial infection.

**Abstract:**

This study investigated the effects of diet with different amylose–amylopectin ratios (AAR) on the growth performance, intestinal morphology, digestive enzyme activities and mRNA expression of nutrients transporters in piglets with short-term lipopolysaccharide (LPS) intraperitoneal injections. Sixty 21 days-old piglets (Landrace × Yorkshire; 6.504 ± 0.079) were randomly assigned based on their body weight (BW) and litters of origins to five groups with experimental diets with an AAR of 0.00, 0.20, 0.40, 0.60, or 0.80 (namely, the 0.00, 0.20, 0.40, and 0.80 groups), respectively. Each treatment included 12 piglets (one piglet per pen). This experiment lasted for 28 days. On the 28th day, six piglets in each treatment were randomly selected for an LPS intraperitoneal injection (100 μg/kg BW), and other piglets were injected with normal saline. Twelve hours after LPS injection, all piglets were sacrificed to collect small intestinal mucosa for analysis. Although different AAR did not influence the final BW in piglets, the piglets in the 0.40 group represented the poorest feed-to-gain ratio (F/G) in the first, second and fourth week (*p* < 0.05) and the lowest average daily gain (ADG) in the fourth week (*p* < 0.05) compared with other groups. In terms of the small intestinal morphology, piglets in the 0.20 and 0.60 groups showed better ileal villous width (*p* < 0.05). Piglets in the 0.60 group presented greater activities of jejunal maltase, sucrase and alkaline phosphatase (*p* < 0.05) than those of 0.20 and 0.40. However, a low amylose diet increased the mRNA expression of jejunal glucose and amino acid transporters (*p* < 0.05). In addition, compared to saline injection, the LPS challenge significantly lessened jejunal digestive enzyme activities (*p* < 0.01) and, ileal villous width and downregulated the gene expression of glucose and amino acid transporters (*p* < 0.05) in piglets. Interestingly, certain diet -LPS interactions on duodenal VH/CD, jejunal maltase activity (*p* < 0.05) and the expression of glucose transporters (*p* < 0.05) were observed. Taken together, in terms of small intestinal digestion and absorption capacity, these results demonstrated that a diet with an AAR of 0.60 diets could improve the intestinal digestive and absorptive capability by affecting small intestinal morphology, digestive enzymes, and nutrients absorptions in piglets. In addition, the diets containing an AAR of 0.40–0.60 were more likely to resist the damage of LPS stress to intestinal morphology and nutrient absorption.

## 1. Introduction

Starch is the primary carbohydrate source in most mixed diets and the major energy source for most animals. It is a mixture of amylose and amylopectin based on the chemical structure of its skeleton chains, it is also deemed to be digestible and resistant starch (RS) due to its digestible potential, and a high level of amylose is associated with highly RS [1]. Starch usually comprises 70–80% of amylopectin and 20–30% of amylose, waxy starch contains less than 1% of amylose, and high amylose starch contains more than 70% of amylose [2]. The starch digestion rate was proved important in pig nutrition because of its potential effect on the plasma insulin level and utilization efficiency of dietary protein [3].

It is well-known that weaning is significant for the life of pigs [4,5]. Evidence has shown that piglets are weaned in an unhealthy status, which will lead to intestinal dysfunction and finally cause physiological disorders [6,7]. Under such circumstances, it is extremely important to prevent and intervene in these disorders by changing dietary composition. The starch ingredient occupies the largest proportion in the diet composition [7]. It is of great significance to improve the use of starch for optimizing the amylose–amylopectin ratio (AAR) in the dietary. The application effect of dietary starch in pigs varies with the difference in starch composition [8,9,10]. In addition, piglets were reported to have a limited influence on the utilization of barley starch in starch composition [8]. To achieve the maximum fractional synthesis rate of proteins in viscera tissues, Deng et al. [9] found that an AAR of 0.23 brought about the optimal results for pig production [9]. Moreover, a diet with an AAR of 0.25 could enhance the level of intestinal digestive enzymes in juvenile obscure puffers compared to a lower or higher amylose diet [11]. Compared with the supplementation of lower amylose, the starch containing 700 g/kg of amylose increased the feed intake and weight gain by 9.1% and 10.6%, respectively [12]. Dietary AAR also exerted an impact on intestinal morphological structure and function. In addition, weaned piglets fed diets containing 160 g/kg of raw potato starch (rich in RS) represented an improved villous height (VH) compared to those fed diets with less raw potato starch [13]. Although studies have indicated the effect of slowly digestible starch on the mRNA expression of nutrient transporters in the ileum compared to the rapidly digestible starch [14], systematic studies on the effects of starch structure on intestinal morphology, digestion, and absorption of piglets are still limited.

In addition, lipopolysaccharide (LPS) is a component of the cell wall of gram-negative bacteria and shows efficacy in highly efficient proinflammation, which has been widely used to model bacterial infection experimentally in animals [15]. RS could improve the intestinal function of weaned pigs owing to its prebiotic properties [16]. Klingbeil et al. [17] identified that a higher serum LPS content induced by the high-fat diet could be reversed by the supplement of RS. An acute stress model of piglets was used to investigate the mechanism that AAR in the diet could strengthen the tolerance of the small intestine to the acute stress from LPS (LPS was injected intraperitoneally 12 hours before sampling). We speculated that a diet containing high amylose could improve intestinal health when weaned piglets underwent the feed transition. This study determined the dietary effects of different AARs on the growth performance, intestinal morphological structures, digestive enzyme activities and mRNA abundance of glucose and amino acids transporters of LPS-challenged and non-LPS challenged weaned piglets.

## 2. Materials and Methods

The experimental procedures in our research were reviewed and approved by the Animal Care and Use Committee of Hunan Normal University, Changsha, Hunan, China.

### 2.1. Experimental Design and Dietary Treatments

Sixty piglets (Landrace × Yorkshire), weaned at 21 days, were randomly assigned based on weight and litter of origin to five groups with diets (every diet contained 12 piglets) with different AARs of 0.00, 0.20, 0.40, 0.60, and 0.80, respectively. Namely, the five groups were 0.00, 0.20, 0.40, 0.60, and 0.80 groups. Diets were formulated to meet or exceed the National Research Council nutrient specifications for weaned piglets [18] and provided in a two-phase feeding program (nursing diets: day 1 to day 14; weaned diets: day 15 to day 29). The ingredients and nutrient levels of the diets could be obtained in another published paper [19]. The differences in dietary AAR were induced by the application of various using different waxy corn starch (Fuyang Biological Starch Co. Ltd., Dezhou, Shandong, China) and high maize (National Starch and Chemical Company, Shanghai, China). Piglets were individually housed during the 28 days experimental periods. At the end of the feeding, 12 hours before sacrifice, six piglets were randomly selected from each group for an intraperitoneal injection of LPS (from Escherichia coli O55:B5, Sigma Chemical Inc., St Louis, MO, USA, L2880), and the other piglets were injected with an equal amount of sterile saline. LPS was dissolved into sterile saline and administered intraperitoneally at 100 μg/kg body weight (BW).

All piglets had free access to feed and water throughout the experimental period. BW was recorded weekly, and daily feed intake was recorded during the experimental period so as to calculate average daily gain (ADG), average daily feed intake (ADFI), and feed-to-gain ratios (F/G). We evaluated the growth performance based on ADG, ADFI, and F/G using the method of Zong et al. [20].

### 2.2. Sample Collection and Treatment

At the end of the experiment, all piglets were sacrificed via electrical stunning followed by exsanguination. Then, the small intestinal tissues were collected. Briefly, approximately 20 cm of anterior duodenal, middle jejunal, and distal ileal segments were isolated and then flushed with 0.9% sodium chloride. Each segment was divided into two sections: one was about 2 cm in length and was fixed using a 10% formaldehyde-phosphate buffer, and the other used for mucosal sampling. The mucosal samples were frozen immediately in liquid nitrogen after collection and stored at −80 ℃ until further analysis of enzymes and mRNA expression was required.

### 2.3. Intestinal Morphological Analysis

Formalin-fixed small intestine samples were embedded in paraffin wax using standard paraffin-embedding techniques. Sections were cut into 4-μm thickness using a microtome (RM2235; Leica, Germany) and were stained with hematoxylin and eosin. Intestinal VH, villous width (VW), crypt depth (CD) and villous surface area (VSA) were observed under a light microscope (DM3000; Leica) at 10×-combined magnification and measured on an Image-ProPlus 6.0 software (Media Cybernetics, San Diego, CA, USA). The average values of VH and CD were calculated from 30 well-oriented, intact villi and their associated crypts [21,22].

### 2.4. Intestinal Enzyme Activity Analysis

Jejunal mucosa tissues were homogenized in 0.9% NaCl solution and centrifuged at 3000× *g* (10 min, 4 ℃). Maltase, sucrase, lactase, and alkaline phosphate (ALP) activities were assayed using commercial kits (Nanjing Jiancheng Bioengineering Institute, Nanjing, China) following the manufacturer’s instructions. The total protein content of each sample was investigated using a protein assay kit (Beyotime Biotechnology, Shanghai; China).

### 2.5. RNA Extraction and cDNA Synthesis

The intestinal mucosa tissue from each sample was pulverized in liquid nitrogen. Total RNA was isolated using Trizol reagent (TaKaRa, Beijing, China), and 100 mg sample per millimeter of Trizol. RNA integrity was detected using a 1% agarose gel electrophoresis and stained with 10 μg/mL ethidium bromide. The quality and quantity of RNA were examined under a UV spectrophotometer (NanoDrop ND-1000; Thermo Fisher Scientific, Waltham, MA, USA). One Cg of RNA (DNA-free) was used for reverse transcription and polymerase. First-strand cDNA was synthesized using an RT Reagent Kit and gDNA eraser (TaKaRa, Beijing, China).

### 2.6. Real-Time PCR Quantification

Primers used for quantification PCR (qPCR) were listed in Table 1. Primers were verified with Primer-BLAST (NCBI). The efficiency and specificity were checked using a melting curve analysis. qPCR was performed under a Step One Plus TM system ((QuantStudio, Thermo Fisher Scientific). Each reaction comprised of 5μL of SYBR Green mix (TaKaRa), 0.2 μL of forward and reverse primers, 0.2 μL of ROX Reference Dye (50×), 3.4 μL of water, and 1 μL of five-fold diluted cDNA [23]. Forty amplification cycles were performed after pre-denaturation (10 s, 95 ℃). Each cycle included 5 s at 95 ℃, 20 s at 60 ℃ and was followed by a melting curve program which was 60 to 99 ℃ with a heating rate of 0.1 ℃/s and fluorescence measurement was carried out. β-actin amplification was used to normalize the target gene expressions for each sample. mRNA relative expression was analyzed using the 2–∆∆Ct method [23]. Real-time reverse-transcription PCR efficiency was determined with by amplifying a cDNA dilution series via equation 10(−1/slope). Target mRNA and β-actin mRNA were amplified with comparable efficiencies. cDNA was replaced by water in the negative controls [22].

### 2.7. Statistical Analyses

Growth performance data (*n* = 59) were analyzed with a one-way ANOVA using SPSS software (version 20.0; IBM Corp., Chicago, IL, USA). The effects between dietary treatment and LPS were evaluated univariately in a normal mixed-linear model using the SPSS General Linear Model procedure (*n* = 56). In addition, all the data were analyzed by regression (curve estimation) to detect the effects of different AARs on the growth performance, intestinal morphology, digestive enzyme activities, and nutrient transporters expression of the piglets. Statistical difference was significant at *p* < 0.05, and a trend toward significance at *p* < 0.10.

## 3. Results

### 3.1. Growth Performance

Four piglets died, among which one was from the 0.00 group, and died in the first week, while the other three pigs died after the LPS challenge from the 0.20, 0.60, and 0.80 groups, respectively. The growth performance of piglets in response to different dietary AARs is summarized in Table 2. As can be seen, final BW and ADFI had no significant differences among the five groups. However, in the first week (WK1), ADG showed a linearly decreasing trend (linear, 0.05 < *p* < 0.1), and the F/G (total, *p* < 0.05; linear, *p* < 0.1) was poorer in the 0.40 and 0.80 AAR groups than in other groups. In addition, with the increase in amylose, ADG was improved significantly while the F/G was lessened significantly (linear and quadratic, *p* < 0.05) during the second week. The growth performance of piglets with dietary of different AARs showed no remarkable alteration in the third week (WK3). In the fourth week, ADG (total, *p* < 0.05) and F/G (total, *p* < 0.05) presented the poorest level in the 0.40 group. Other treatments had no differences.

### 3.2. Intestinal Morphology

As shown in Table 3, the 0.20, 0.40, and 0.60 groups presented greater duodenal VW (AAR, *p* < 0.05; quadratic, *p* < 0.01), and the 0.60 group also had a greater ileal VW than the other groups (AAR, *p* < 0.05). In addition, the LPS challenge significantly decreased the ileal VW and VSA of piglets (LPS, *p* < 0.01), and a decreasing trend was observed for the jejunal CD (LPS, 0.05 < *p* < 0.1) and ileal VSA (LPS, *p* = 0.05). In addition, an interaction was observed between dietary AAR and LPS in duodenal VH/CD (AAR × LPS, *p* < 0.05), revealing that piglets in the 0.60 group represented the strongest resistance to the LPS challenge.

### 3.3. Jejunal Digestive Enzyme Activities

As shown in Table 4, the jejunal maltase activity varied linearly and was quadratic with the increasing amylose (AAR, linear, quadratic, *p* < 0.05). Nevertheless, the sucrase (AAR, *p* < 0.05) activity of the piglets in the 0.60 group was greater than that in other groups. Additionally, the activities of maltase, sucrase and alkaline phosphatase activity (LPS, *p* < 0.01) were lower in the LPS-challenged piglets than those in the non-challenged piglets. Some significant interactions between dietary AAR and LPS (AAR × LPS, *p* < 0.05) were observed in the activities of maltase and lactase, and the piglets in the 0.40–0.60 groups showed more resistance to LPS-induced change of maltase activity.

### 3.4. mRNA Relative Expression of Intestinal Nutrient Transporters

The mRNA relative expression values of nutrient transporter are presented in Table 5. As can be seen, the jejunal expression of Slc2a2 (Linear, *p* < 0.05), Slc6a19 (AAR, *p* < 0.01; linear, *p* < 0.05) and Slc7a1 (AAR, linear and quadratic, *p* < 0.05) decreased linearly. The jejunal expression of Slc7a1 (LPS, *p* < 0.05) significantly decreased in the LPS-challenged piglets.

In the ileum, diets with different AARs could significantly affect the expression level of Slc5a1 (Table 5; AAR, *p* < 0.001), Slc2a2 (AAR, *p* < 0.05), Slc1a1 (linear and quadratic, *p* < 0.01), Slc6a19 (linear and quadratic, *p* < 0.01), Slc7a1 (AAR, *p* < 0.01) and Slc7a9 (AAR, *p* < 0.05; linear, *p*< 0.01), which might imply that low amylose dietary might promote the absorption of ileal glucose, but the high amylose may save the absorption of glucose and amino acids. In addition, the intraperitoneal injection of LPS significantly reduced the expression of Slc5a1 (LPS, *p* < 0.01) and Slc2a2 (LPS, *p* < 0.01) in piglets.

In addition, certain remarkable interaction effects of AAR and LPS were found on jejunal Slc2a2 and Slc7a1, and ileal Slc5a1 (AAR × LPS; *p* < 0.05), indicating that the 0.40 group was significantly increased in response to the LPS challenge.

## 4. Discussion

Starch is a primary source of carbohydrates and energy for animals and plays an important role in animal growth. Numerous studies have identified that diets of different starch sources can affect animal growth [11,24]. However, the relevance of AAR to the intestinal digestive and absorptive capability in weaned piglets under LPS stress remains unknown. This research can provide guidance for the nutritional intervention so as to ameliorate weaning stress or bacterial infection of piglets.

Our data showed that dietary AAR did not affect ADFI but ADG and F/G. Furthermore, the ADG and F/G in piglets in the 0.40 group were the worst compared with other groups in the fourth week, which may be attributed to the poorer digestive and absorptive capability proven by the lower digestive enzyme activities and nutrient transporters caused by severe diarrhea in the third week. The data could be available in our other paper [19]. Additionally, in the second week, the 0.60–0.80 AAR groups represented significantly improved F/G, which, however, disappeared in the fourth week. These results demonstrated that the diet with higher amylose could improve F/G in a short time (two weeks), which might imply that starch fermentation in the small intestine was increased, and the end fermentation products were beneficial to piglets [25]. Furthermore, similar results could also be observed in the study of Li et al. [26], that higher amylose diets significantly improved ADG and F/G but did not alter the ADFI in growing pigs. However, ADFI showed no change with elevated amylose concentrations, consistent with the study of van Erp et al. [25].

The morphological structures of the small intestine are closely associated with the intestinal digestive and absorptive capabilities. Significant changes, such as villous atrophy, crypt hyperplasia, and declined enzyme activities, occurred in the intestinal structures and functions of weaned piglets, generally induced by less growth performance [11]. This work revealed that diets with AARs of 0.20 and 0.60 improved small intestinal morphological structures. Greater VW and VSA in the 0.20 and 0.60 groups could increase the digestive and absorptive surface area of the small intestine, and similar results were observed in chicken [12], and piglets in another study [13]. Additionally, the LPS challenges remarkably reduced the jejunal VH and ileal VH/CD due to decreased cell renewal rates or increased cell losses [27].

As is well-known, starch is digested into disaccharides in the small intestine, or fermented by intestinal microorganisms in the large intestine [28]. Adeola and King [29] reported that the development of the intestinal digestive enzyme activity was an important factor in estimating the capabilities of growing pigs for macromolecules digestion. The present study showed that a diet with different AARs could significantly affect the jejunal digestive enzyme activities. A previous study indicated that digestive enzyme activities were relevant to VH [30]. Interestingly, our result on digestive enzyme activities showed that a diet with an AAR of 0.60 significantly enhanced jejunal disaccharidase activities compared with that in other groups. We speculated this ratio in the diet increased the starch left in the small intestine after 12 h fasting and capable of being utilized by the small intestine, and more digestive enzymes were required to digest diets fully, resulting in less undigested starch flows into the large intestine to be utilized by microorganisms, compared with other groups in the current study, and thus reducing ameliorated the utilization of starch. This can be verified by the published data on short-chain fatty acids in the cecum [19]. In addition, we supposed that an AAR of 0.80 contained more slowly digestive starch, which might exceed the digestive capacity of the small intestine. However, these differences were not observed in the study of Gao et al. [7], which may be caused by the inconsistency of used starch types. We used waxy corn starch and high maize, while the purified maize starch was selected by Gao et al. [7]. Additionally, the decreased jejunal digestive enzyme and ileal glucose transporters in LPS-challenged piglets might be related to the reduction of jejunal VH [31].

The absorption of dietary sugar and amino acids is primarily mediated by specific transporters [14,23]. Slc5a1 and Slc2a2 are primarily glucose transporters expressed on piglet enterocyte membranes [20]. Slc1a1, Slc6a19, Slc7a1 and Slc7a9 have been recognized as the major intestinal transporters for acidic, neutral, and basic amino acids [23,32]. According to the study of van Kempen et al. [33], the in vivo glucose release affected nutrient absorption kinetics. Woodward et al. investigated that slowly digestible starch increased intestinal glucose transporters, possibly due to a shift in substrate availability [14]. Similar to glucose transport, amino acid absorption also needs coupled transport with Na+ absorption, and thus the absorption of intestinal glucose and amino acids might compete with amino acids [34]. The continuous supply of glucose in the intestinal lumen could increase glucose used directly, improving the efficiency of glucose utilization, and thus, reducing the use of amino acids for oxidative energy supply. As Slc5a1 expression could be affected by certain amino acids, high maize starch was influenced by the net portal flux of amino acids by impacting glucose in the portal vein [32,35]. In our study, dietary AAR also affected the expression of glucose and amino acid transporters in the small intestine. Moreover, piglets fed the diet with lower amylose (0.00–0.20) represented a significant upregulation of the expression of amino acid transporters in the jejunum and glucose transporters in the ileum. This showed that rapidly digestive starch produced more glucose to be absorbed by the body simultaneously, which could be proved by the experiment of Woodward et al. [14]. A diet with an AAR 0.60 significantly decreased amino acid transporters expression in the small intestine, we supposed this may imply a saving of amino acid and improved glucose utilization. However, a diet with an AAR of 0.80 probably contained more indigestible starch, thereby reducing glucose absorption and requiring more amino acids to maintain energy needs.

In addition, some interactions were found between AARs and LPS, and the results showed that diets with 0.40–0.60 of AAR could resist the changes induced by of LPS-challenge. This might be attributed to the early adaptation to acute stress because this group experienced severe diarrhea in the third week, which could be proven by the increasing cecal short-chain fatty acids, and intestinal microbes [19]. However, Gao et al. [7] had shown that high amylose (AAR 2.90) could influence intestinal health by improving the intestinal barrier functions and reducing enterocyte apoptosis, which was different from our results. We deemed the amylose content far beyond the basal diet, which might not be beneficial for long-term feeding. The author also acknowledged that the high amylose levels were detrimental to piglet absorption. In addition, the other research indicated that RS had positive effects on gut intestinal health [12,16,17], basically consistent with our results, the AAR of 0.40–0.60 could improve the unideal intestinal health caused by LPS stress, while the higher amylose content might cause malabsorption in pigs.

## 5. Conclusions

Taken together, the diet with an AAR of 0.60 may improve small intestinal morphology, and nutrient digestion and save amino acid utilization, thereby improving the feed utilization of piglets. In addition, piglets fed diets with an 0.40 or 0.60 AAR possibly had healthier small intestines in response to the LPS challenge, including intestinal morphology, digestive enzyme activity and nutrient transporters. These results are expected to provide a way for developing different AAR as a functional food additive for piglets.

## Figures and Tables

**Table 1 animals-12-01833-t001:** Primers used for real-time PCR analysis.

Genes	Primers	Sequences (5’-3’)	Size bp	GeneBank Accession No.
*Slc2a2*	Forward	AAGTCGAGGCCTATGATCTGACTAA	161	NM_001097417.1
	Reverse	GGAAGAGGCATATCAGGACTCTACT		
*Slc5a1*	Forward	ATCTCTGTCATCGTCATCTAC	121	NM_001164021.1
	Reverse	GCCACCACACCATACTTC		
*Slc1a1*	Forward	GCTGTGCTGAAGAGAAGAA	181	NM_001164649.1
	Reverse	GTGGCGGTGATACTGATAG		
*Slc6a19*	Forward	CACAACAACTGCGAGAAG	152	XM_003359855.4
	Reverse	TTGATAAGCGTCAGGATGT		
*Slc7a1*	Forward	CCCCTGTGGTAGCGATGCAGTCA	229	NM_001012613.1
	Reverse	CTGGGCTTCATAATGGTGTCAGGAT		
*Slc7a9*	Forward	GAAGAAGCCTCCTAGAAGTG	268	NM_001110171.1
	Reverse	CCAGTGTCGCAAGAATCC		
*β-actin*	Forward	AGTTGAAGGTGGTCTCGTGG	216	XM_003357928.4
	Reverse	TGCGGGACATCAAGGAGAAG		

**Table 2 animals-12-01833-t002:** Influence of diet with different amylose–amylopectin ratio on the growth performance of weaned piglets ^1^.

Item ^2^	AAR					SEM	*p*-Value		
	0.00	0.20	0.40	0.60	0.80		Total	Linear	Quadratic
ADFI, g									
WK1	273.53	305.70	273.08	254.95	278.23	8.87	0.52	0.47	0.77
WK2	354.31	384.30	357.45	378.60	412.95	16.76	0.82	0.36	0.60
WK3	484.41	420.11	474.97	472.03	496.07	18.29	0.76	0.72	0.68
WK4	675.84	705.67	630.80	620.64	653.11	20.01	0.68	0.20	0.30
Total	502.34	475.50	464.63	453.39	480.58	15.01	0.90	0.44	0.41
ADG, g									
WK1	117.86	143.45	92.21	105.36	96.10	6.27	0.05	0.09	0.24
WK2	148.75	122.22	161.00	186.36	233.00	13.18	0.08	0.02	0.03
WK3	239.38	197.92	247.40	198.75	180.73	13.33	0.44	0.32	0.85
WK4	445.71 ^a^	374.40 ^ab^	286.36 ^b^	420.13 ^a^	366.67 ^ab^	15.54	0.01	0.34	0.10
Total	236.48	206.79	203.24	223.57	210.96	8.80	0.78	0.35	0.30
F/G									
WK1	2.44 ^ab^	1.95 ^b^	3.00 ^a^	2.43 ^ab^	2.79 ^a^	0.11	0.02	0.09	0.25
WK2	2.70 ^a^	2.71 ^a^	2.71 ^a^	2.01 ^b^	1.92 ^b^	0.12	0.01	0.01	0.02
WK3	2.21	2.21	1.96	2.51	2.54	0.10	0.30	0.09	0.09
WK4	1.84 ^b^	2.07 ^b^	2.45 ^a^	1.92 ^b^	2.04 ^b^	0.06	0.03	0.82	0.20
Total	2.11	2.36	2.36	2.16	2.37	0.05	0.24	0.45	0.63
Initial BW, g	6.63	6.46	6.56	6.42	6.45	0.08	0.92	0.88	0.44
Final BW, g	12.50	12.05	12.05	12.04	12.15	0.28	0.99	0.74	0.85

^1^ A total of 56 piglets. ^2^ WK = week; BW = body weight; ADFI = average daily feed intake; ADG = average daily gain; F/G: feed to gain ratio; AAR = amylose–amylopectin ratio; SEM = standard error of the mean. ^a,b^ Values within a row with different superscripts differ significantly at *p* < 0.05.

**Table 3 animals-12-01833-t003:** Influence of diet with different amylose–amylopectin ratio and LPS challenge (intraperitoneal injection) on the intestinal morphology of weaned piglets ^1^.

Item ^2^	0.00		0.20		0.40		0.60		0.80		SEM	*p*-Value				
	LPS	SAL	LPS	SAL	LPS	SAL	LPS	SAL	LPS	SAL		AAR	LPS	AAR × LPS	L	Q
Duodenum																
VH, μm	314.4	375	374.2	337.2	295.7	312.7	327.3	311.9	291.4	330.6	7.05	0.11	0.34	0.17	0.09	0.23
VW, μm	117.3 ^b^	117.5	124.8 ^ab^	132.2	131 ^a^	124.3	125.5 ^ab^	135.6	116.3 ^b^	119.2	1.65	0.02	0.54	0.53	0.81	<0.01
CD, μm	412.7	424.1	360.1	431.3	421.1	357.2	419.9	428	382.7	356.5	8.96	0.16	0.79	0.12	0.12	0.28
VH/CD	0.79	0.9	1.07	0.79	0.71	0.9	0.8	0.75	0.78	1.00	0.03	0.41	0.41	0.04	0.77	0.76
VSA, mm^2^	0.12	0.14	0.15	0.14	0.12	0.12	0.13	0.13	0.11	0.13	<0.01	0.06	0.23	0.68	0.14	0.13
Jejunum																
VH, μm	295.3	341.3	321.6	344.5	299.6	315.8	344	344.6	318	310.2	5.11	0.17	0.14	0.48	0.78	0.63
VW, μm	108.4	123.5	119.1	124.9	117.2	116.2	119.4	114.7	110.1	109.1	1.64	0.28	0.4	0.31	0.12	0.06
CD, μm	250.3	315.9	264	279.3	276.6	283.3	281.4	293.5	277.3	282.1	5.61	0.94	0.07	0.35	0.68	0.91
VH/CD	1.2	1.12	1.22	1.24	1.11	1.13	1.24	1.19	1.15	1.12	0.03	0.2	0.12	0.15	0.52	0.21
VSA, mm^2^	0.1	0.13	0.12	0.14	0.11	0.12	0.13	0.12	0.11	0.11	<0.01	0.77	0.68	0.98	0.53	0.36
Ileum																
VH, μm	264.6	250.3	256.7	290.5	238.2	275.9	265.9	262.7	225	252.9	5.76	0.43	0.16	0.51	0.23	0.25
VW, μm	108.7 ^b^	116.2	113.2 ^ab^	126.3	106.1 ^b^	115.5	116.2 ^a^	141.8	113 ^ab^	116.6	1.6	0.01	<0.01	0.26	0.3	0.36
CD, μm	252.2	260.6	264.3	252	256.9	252	255.1	233.8	247.8	253.8	5.04	0.92	0.64	0.87	0.49	0.79
VH/CD	1.05	0.96	0.98	1.16	0.93	1.12	1.06	1.14	0.9	1	0.02	0.17	<0.01	0.82	0.56	0.28
VSA, mm^2^	0.09	0.1	0.09	0.11	0.08	0.1	0.1	0.12	0.08	0.09	<0.01	0.29	0.05	0.34	0.65	0.47

^1^ A total of 56 piglets. ^2^ VH = villous height; VW = villous width; CD = crypt depth; VSA = villous surface area; AAR = amylose–amylopectin ratio; LPS = lipopolysaccharide; SAL= saline; L = linear; Q = Quadratic; SEM = standard error of the mean. ^a,b^ Values within a row with different superscripts differ significantly at *p* < 0.05.

**Table 4 animals-12-01833-t004:** Influence of diet with different amylose–amylopectin ratio and LPS challenge (intraperitoneal injection) on the jejunal digestive enzyme activities of weaned piglets ^1^.

Item ^2^	0.00		0.20		0.40		0.60		0.80		SEM	*p*-Value				
	LPS	SAL	LPS	SAL	LPS	SAL	LPS	SAL	LPS	SAL		AAR	LPS	AAR × LPS	L	Q
Maltase, U/mgprot	144.3 ^bc^	236.9	88.7 ^c^	166.4	217.4 ^b^	199.1	281.6 ^a^	289.9	141.0 ^bc^	312.4	9.7	<0.01	<0.01	0.03	<0.01	0.02
Sucrase, U/mgprot	16.3 ^ab^	30.2	11.4 ^b^	25.9	11.7 ^b^	20.2	25.8 ^a^	33.9	13.4 ^b^	19.2	1.3	0.01	<0.01	0.81	0.83	0.88
Lactase, U/gprot	9.9	15.7	6.1	5.1	6.4	6.0	10.4	10.9	10.9	11.9	0.9	0.07	0.11	0.59	0.51	0.04
ALP, U/mgprot	108.7	151.7	56.5	143.8	74.1	117.3	62.7	99.2	70.6	170.3	6.3	0.14	<0.01	0.39	0.56	0.53

^1^ A total of 56 piglets. ^2^ ALP = Alkaline phosphatase; AAR = amylose–amylopectin ratio; LPS = lipopolysaccharide; SAL = saline; L = linear; Q = quadratic; SEM = standard error of the mean. ^a–c^ Values within a row with different superscripts differ significantly at *p* < 0.05.

**Table 5 animals-12-01833-t005:** Influence of diet with different amylose–amylopectin ratio and LPS challenge (intraperitoneal injection) on the nutrient transporters in jejunum and ileum of weaned piglets ^1^.

Item ^2^	0.00		0.20		0.40		0.60		0.80		SEM	*p*-Value				
	LPS	SAL	LPS	SAL	LPS	SAL	LPS	SAL	LPS	SAL		AAR	LPS	AAR × LPS	L	Q
Jejunum																
*Slc5a1*	1.00	1.09	0.96	1.09	0.81	0.69	0.69	0.78	0.49	1.13	0.08	0.52	0.27	0.61	0.10	0.20
*Slc2a2*	0.93	1.05	1.11	1.11	0.98	0.59	0.58	0.94	0.70	0.96	0.04	0.07	0.42	0.04	0.04	0.10
*Slc1a1*	0.87	1.13	1.28	1.24	0.96	1.19	1.39	0.82	0.85	1.41	0.07	0.83	0.54	0.14	0.95	0.89
*Slc6a19*	0.92 ^a^	1.04	1.21 ^a^	0.75	1.11 ^a^	0.85	0.59 ^b^	0.52	0.79 ^a,b^	0.80	0.04	<0.01	0.13	0.25	0.02	0.05
*Slc7a1*	0.95 ^a,b^	1.05	1.18 ^a^	0.29	0.96 ^a,b^	0.77	0.55 ^b^	0.42	0.62 ^b^	0.70	0.05	0.02	0.04	0.03	0.01	0.04
*Slc7a9*	0.88	1.08	0.89	0.85	0.98	0.98	0.85	0.73	0.62	0.90	0.05	0.43	0.51	0.66	0.08	0.18
Ileum																
*Slc5a1*	1.03 ^a^	0.93	0.80 ^a^	1.57	0.69 ^b^	0.53	0.53 ^b^	0.72	0.36 ^c^	0.56	0.04	<0.01	0.02	<0.01	0.12	0.08
*Slc2a2*	1.09 ^a^	1.75	0.63 ^a,b^	0.99	0.37 ^b^	0.92	0.54 ^a,b^	0.98	0.86 ^a,b^	1.00	0.07	0.01	<0.01	0.81	0.07	0.17
*Slc1a1*	1.03	0.88	0.86	1.63	0.79	0.74	0.76	1.06	0.78	0.88	0.06	0.15	0.10	0.16	<0.01	<0.01
*Slc6a19*	1.01	0.96	0.90	1.38	1.70	1.35	1.28	1.57	1.05	1.52	0.08	0.27	0.33	0.46	<0.01	<0.01
*Slc7a1*	1.10 ^b^	1.10	1.39 ^b^	0.91	2.26 ^a^	1.82	2.52 ^a^	1.99	1.88 ^a,b^	2.22	0.12	<0.01	0.34	0.68	0.26	0.53
*Slc7a9*	1.05 ^a,b^	0.88	1.50 ^a^	1.55	0.89 ^b^	0.96	0.60 ^b^	0.91	0.80 ^b^	0.94	0.07	0.02	0.54	0.84	0.12	<0.01

^1^ A total of 56 piglets. ^2^
*Slc2a2* = glucose transporter 2; *Slc5a1*: sodium glucose linked transporter 1; *Slc1a1* = acidic amino acid transporter; *Slc6a19* = neutral amino acid transporter; Slc7a1, *Slc7a9* = basic amino acid transporter; AAR = amylose–amylopectin ratio; LPS = lipopolysaccharide; SAL = saline; L = linear; Q = quadratic; SEM = standard error of the mean. ^a–c^ Values within a row with different superscripts differ significantly at *p* < 0.05.

## Data Availability

The datasets that were applied and/or analyzed throughout the prevailing research are available from the corresponding author.

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
