# Peer review of "Effects of Dietary Amylose—Amylopectin Ratio on Growth Performance and Intestinal Digestive and Absorptive Function in Weaned Piglet Response to Lipopolysaccharide"

_animals, 2022, doi:10.3390/ani12141833_

Round 1
Reviewer 1 Report
Starch is the most abundant nutrient in rations for farm animals. It consists of amylose and amylopectin, the proportion of which has a significant effect on digestive processes. Against this background, it is extremely interesting that the relationship between these two components is being investigated.
This is done in a study whose design was planned very clearly and stringently. The explanations in the mouse script are also very precise and cleanly formulated. Nevertheless, there are some comments that should now follow:
Line 35 Yorkashire (in line 111 Yorkshire)
Line 65 delete the blank behind potential,
Line 67 ….amylose. Waxy starches contain <
Line 68 contain
Line 72 unhealthy status
Line 72 lead to intestinal…
Line 75 largest ingredients
Line 79 barley starch in barley starch???
Line 81 However, to achieve….
Line 89 lower contens of raw…
Line 97/97 and a higher serum LPS content…. Of RS [15]
Line 125 use a blank between 1ßß and mikrogramm
Line 130 it is not [18]
Line 138 -80 °C
Line 151 at 4 °C
Line 166 veri-fied
Line 172 60 °C
Line 172 99 °C
Line 174 0.1 °C
Line 182 and 192 when 4 piglets died why n=59?
Line 214 the jejunal
Line 223 the design under items is not the best
Line 239 the chapter starts in the middle?
Line 270 with the corresponding data of other…
Line 279 delete in the study of Li;Li
Line 282 Erp et al.
Line 308 Gao et al.
Line 310 dito
Line 318 delte: the study of …until that; kinetics and slowly digestible starch
Line 320 behind availability [12]
Line 323 (van den Borne et al. 2007) ???
Author Response
Dear Reviewer 1
Thank you very much for your and reviewers’ constructive comments concerning our manuscript entitled “Effects of dietary amylose-amylopectin ratio on growth performance and intestinal digestive and absorptive function in weaned piglet response to lipopolysaccharide” (animals-1723162). Those criticisms are all valuable and very helpful for improving our paper. We have carefully revised our manuscript in accordance with your kind advices and reviewers’ detailed comments.
We would like to submit a revision for your consideration again. Looking forward to hearing from you soon.
With best regards,
Sincerely,
Huansheng Yang Ph.D.
e-mail: [email protected]
Comments and Suggestions for Authors
Starch is the most abundant nutrient in rations for farm animals. It consists of amylose and amylopectin, the proportion of which has a significant effect on digestive processes. Against this background, it is extremely interesting that the relationship between these two components is being investigated.
This is done in a study whose design was planned very clearly and stringently. The explanations in the mouse script are also very precise and cleanly formulated. Nevertheless, there are some comments that should now follow:
Line 35 Yorkashire (in line 111 Yorkshire)
Response: Thank you very much for carefully inspection. We had revised in line 36 and 111.
Line 65 delete the blank behind potential,
Response: Thank you very much for carefully inspection. We had delete the blank in Line 65.
Line 67 amylose. Waxy starches contain <
Response: Thank you very much for carefully observation. We had revised it in Line 67.
Line 68 contain
Response: Thank you very much for carefully inspection. We had revised it in Line 67.
Line 72 unhealthy status
Response: Thank you very much for your sincere comment. We had revised it in Line 72.
Line 72 lead to intestinal…
Response: Thank you very much for carefully inspection. We had revised it in Line 72.
Line 75 largest ingredients
Response: Thank you for your careful observation. We had revised it in line 75.
Line 79 barley starch in barley starch???
Response: Thank you for your sincere question. We had deleted “barley” in line 79.
Line 81 However, to achieve….
Response: Thank you for your careful suggestion. We had revised them in line 81-82.
Line 89 lower contents of raw…
Response: Thank you for your careful observation. We had revised it in line 89.
Line 97/97 and a higher serum LPS content…. Of RS [15]
Response: Thank you for your sincere suggestion. We had revised it in line 98.
Line 125 use a blank between 1ßß and mikrogramm
Response: Thank you for your careful observation. We had revised it in line 125.
Line 130 it is not [18]
Response: Thank you for your sincere question. We cite [18] according to the study of (Zong et al., 2018), but we can’t find [18] now, so we modify it to (Zong et al., 2018) in this paper.
Line 138 -80 °C
Response: Thank you for your careful observation. We had revised it in line 138.
Line 151 at 4 °C
Response: Thank you for your careful comment. We had revised it in line 151.
Line 166 veri-fied
Response: Thank you for your sincere question. We had revised it in line 167.
Line 172 60 °C
Response: Thank you for your careful comment. We had revised it in line 173.
Line 172 99 °C
Response: Thank you for your careful observation. We had revised it in line 173.
Line 174 0.1 °C
Response: Thank you for your careful observation. We had revised it in line 174.
Line 182 and 192 when 4 piglets died why n=59?
Response: Thank you for your sincere question. We started the experiment with 60 piglets. One of them died in the first week of the experiment, and the other three died from LPS after being weighed for the final body weight, so the data for growth performance had 59 piglets.
Line 214 the jejunal
Response: Thank you for your careful observation. We had revised it in line 214.
Line 223 the design under items is not the best
Response: Thank you for your sincere suggestion. We had re-designed them in line 222.
Line 239 the chapter starts in the middle?
Response: Thank you for your sincere question. This chater does not begin in the middle, there is 3.3 ahead.
Line 270 with the corresponding data of other…
Response: Thank you for your sincere question. We believed that digestive and absorptive capability are related to the digestive enzyme activity and nutrient transporter, the corresponding data we can obtain in table 4 and 5. This view is consistent with the previous study (Zong et al., 2018).
Line 279 delete in the study of Li;Li
Response: Thank you for your careful observation. We had revised it in line 278.
Line 282 Erp et al.
Response: Thank you for your careful observation. We had revised it in line 283.
Line 308 Gao et al.
Response: Thank you for your careful observation. We had revised it in line 310.
Line 310 dito
Response: Thank you for your careful observation. We had revised it in line 312.
Line 318 delte: the study of …until that; kinetics and slowly digestible starch
Response: Thank you for your sincere suggestion. We had revised it in line 317-319
Line 320 behind availability [12]
Response: Thank you for your sincere suggestion. We had revised it in line 321.
Line 323 (van den Borne et al. 2007) ???
Response: Thank you for your sincere question. We are so sorry for our carelessness. We had revised it in line 324, 327 and 442-445.
Reviewer 2 Report
Simple summary
Without stating the purpose, suddenly shifting to the results is not scientific. Please add to your summary.
Abstract:
L32: This present study was conducted to investigate ….>>> This study investigated the …
L35: The specification of BW of pigs are required.
L53-57: It is a long sentence for the conclusion. Please break it into two sentences.
Introduction
L71-73: Please rephrase the sentence: “Evidence shows that they are usually in an unhealthy when weaning, which will lead to the intestinal dysfunction and finally rose physiological disorders”
L75-76: This sentence: “To our knowledge, starch is the large ingredient in the diet composition [5]”: It should not be the author's knowledge. It is a practical and scientific fact. Please re-phrase
L77-78: The physiological effect of what? The object is missing.
L81: The citation is not formatted well.
L98: The citation is not formatted well.
M&M
Section and subheadings should be all capitalized. Please follow the format of the “Animals” strictly. Also, p values should be italicized and small all over the text.
Discussion
L267-268: Please re-phrase.
L279: The citation is not formatted well.
L282: The citation is not formatted well.
L310: The citation is not formatted well.
L319: The citation is not formatted well.
L323: This citation does not exist in the reference list. Also, the citation is not formatted well.
L333: The citation is not formatted well.
L344: The citation is not formatted well.
References
Not formatted based on the journal guideline
Author Response
Dear Reviewer 2
Thank you very much for your and reviewers’ constructive comments concerning our manuscript entitled “Effects of dietary amylose-amylopectin ratio on growth performance and intestinal digestive and absorptive function in weaned piglet response to lipopolysaccharide” (animals-1723162). Those criticisms are all valuable and very helpful for improving our paper. We have carefully revised our manuscript in accordance with your kind advices and reviewers’ detailed comments.
We would like to submit a revision for your consideration again. Looking forward to hearing from you soon.
With best regards,
Sincerely,
Huansheng Yang Ph.D.
e-mail: [email protected]
Comments and Suggestions for Authors
Simple summary
Without stating the purpose, suddenly shifting to the results is not scientific. Please add to your summary.
Response: Thank you for your sincere suggestion. We had revised it in simple summary.
Abstract:
L32: This present study was conducted to investigate ….>>> This study investigated the …
Response: Thank you for your sincere suggestion. We had revised it in line 33.
L35: The specification of BW of pigs are required.
Response: Thank you for your carefully suggestion. We had added the data of BW in line 36.
L53-57: It is a long sentence for the conclusion. Please break it into two sentences.
Response: Thank you for your carefully observation. We had revised in line 53-57
Introduction
L71-73: Please rephrase the sentence: “Evidence shows that they are usually in an unhealthy when weaning, which will lead to the intestinal dysfunction and finally rose physiological disorders”
Response: Thank you for your sincere suggestion. We had revised it in line 72-74.
L75-76: This sentence: “To our knowledge, starch is the large ingredient in the diet composition [5]”: It should not be the author's knowledge. It is a practical and scientific fact. Please re-phrase
Response: Thank you for your carefully suggestion. We had revised it in line 76.
L77-78: The physiological effect of what? The object is missing.
Response: Thank you for your sincere suggestion. We apologize for the mis-representation, we had revised it in line 78-79.
L81: The citation is not formatted well.
Response: Thank you for your carefully inspection. We had revised it in line 83.
L98: The citation is not formatted well.
Response: Thank you for your carefully inspection. We had revised it in line 99.
M&M
Section and subheadings should be all capitalized. Please follow the format of the “Animals” strictly. Also, p values should be italicized and small all over the text.
Response: Thank you for your sincere suggestion. We had revised in paper.
Discussion
L267-268: Please re-phrase.
Response: Thank you for your sincere suggestion. We had revised it in line 267-268.
L279: The citation is not formatted well.
Response: Thank you for your carefully inspection. We had revised it in line 279.
L282: The citation is not formatted well.
Response: Thank you for your carefully inspection. We had revised it in line 282.
L310: The citation is not formatted well.
Response: Thank you for your carefully inspection. We had revised it in line 309, 311.
L319: The citation is not formatted well.
Response: Thank you for your carefully inspection. We had revised it in line 319.
L323: This citation does not exist in the reference list. Also, the citation is not formatted well.
Response: Thank you for your sincere inspection. We are so sorry for our carelessness. We had revised it in line 323 and 443.
L333: The citation is not formatted well.
Response: Thank you for your carefully inspection. We had revised it in line 333.
L344: The citation is not formatted well.
Response: Thank you for your carefully inspection. We had revised it in line 343.
References
Not formatted based on the journal guideline
Response: Thank you for your carefully inspection. We had revised it in references.
Reviewer 3 Report
Overall, the paper is well written and shows interesting results, however, some minor revisions need to be addressed as following:
1. With study on intestinal characteristics, it is necessary to know the dietary nutritional composition. Therefore, it is required authors to show the ingredient composition (Table). It's better if
2. Please provide more detail in the method section that how author can formulate different diets with different AAR just based on corn starch and high-maize. How about other ingredients?
3. In method section, total 60 piglets were used for this experiment, however, growth performance (Table 2, n=59) and other table (n=56). Please explain and there may be any relation to the effect of diets?
4. Because diets were formulated according to 2-phase feeding program, it is better for table 2 authors presented the data in 2 phases and overall rather than weekly.
5. It is needed to provide more detail for table 2 in footnote.
6. The present of P value need to be consistent over table.
7. A separate conclusion section needs to be added.
8. L355-357: the conclusion is not supported by results of this study. This study did not test the influence of dietary AARs on the resistance to bacterial infection.
Author Response
Dear Reviewer 3
Thank you very much for your constructive comments concerning our manuscript entitled “Effects of dietary amylose-amylopectin ratio on growth performance and intestinal digestive and absorptive function in weaned piglet response to lipopolysaccharide” (animals-1723162). Those criticisms are all valuable and very helpful for improving our paper. We have carefully revised our manuscript in accordance with your kind advices and reviewers’ detailed comments.
We would like to submit a revision for your consideration again. Looking forward to hearing from you soon.
With best regards,
Sincerely,
Huansheng Yang Ph.D.
e-mail: [email protected]
Comments and Suggestions for Authors
Overall, the paper is well written and shows interesting results, however, some minor revisions need to be addressed as following:
- With study on intestinal characteristics, it is necessary to know the dietary nutritional composition. Therefore, it is required authors to show the ingredient composition (Table). It's better if
Response: Thank you for your sincere suggestion. The ingredient composition we had published in (Yang et al., 2021), so we no longer put it in the article, but we had explained in line 118.
- Please provide more detail in the method section that how author can formulate different diets with different AAR just based on corn starch and high-maize. How about other ingredients?
Response : Thank you for your sincere question. We formulated the different AAR ratios in diets according to different ratio of waxy corn starch (Shandong Fuyang Biological Starch Co.Ltd, Dezhou, Shandong, China) and High-Maize 1043 (National Starch and Chemical Company,Shanghai,China), details can be found in our other article (Yang et al., 2021). And the contents of starch in different diets were 0.04, 0.21, 0.41,0.62 and 0.80, respectively. Since the amount of other ingredients is the same, we did not test the actual value.
- In method section, total 60 piglets were used for this experiment, however, growth performance (Table 2, n=59) and other table (n=56). Please explain and there may be any relation to the effect of diets?
Response: Thank you for your sincere question. During the experiment, One of them died in the first week, and three pigs died less than 12 hours after intraperitoneal of LPS after final weighing, so the data for growth performance had 59 piglets, but the three pigs that died from LPS stress were not sampled, so only 56 samples were analyzed. We don’t think it was related to diet, because the pigs that died were not in the same group.
- Because diets were formulated according to 2-phase feeding program, it is better for table 2 authors presented the data in 2 phases and overall rather than weekly.
Response: Thank you for your sincere comment. We think your suggestion is a good idea, but we also think that the weekly analysis can understand a gradual process (if any) of the effects of different amylose/amylopectin ratio on piglets. And can also get the difference of two stages according to the difference of week.
- It is needed to provide more detail for table 2 in footnote.
Response: Thank you for your carefully inspection. We are so sorry for our carelessness. We had provided the details in Table 2.
- The present of P value need to be consistent over table.
- Response: Thank you for your carefully inspection. We are so sorry for our carelessness. We rechecked all results and their P values and made changes in text.
- A separate conclusion section needs to be added.
Response: Thank you for your sincere suggestion. A separate conclusion section we had provided in line 354.
- L355-357: the conclusion is not supported by results of this study. This study did not test the influence of dietary AARs on the resistance to bacterial infection.
Response: Thank you for your sincere suggestion. We had revised it in line 357-360.
References
Yang, C., M. Wang, X. Tang, H. Yang, F. Li, Y. Wang, J. Li, and Y. Yin. 2021. Effect of Dietary Amylose/Amylopectin Ratio on Intestinal Health and Cecal Microbes' Profiles of Weaned Pigs Undergoing Feed Transition or Challenged With Escherichia coli Lipopolysaccharide. Front Microbiol 12:693839. doi: 10.3389/fmicb.2021.693839
Zong, E., P. Huang, W. Zhang, J. Li, Y. Li, X. Ding, X. Xiong, Y. Yin, and H. Yang. 2018. The effects of dietary sulfur amino acids on growth performance, intestinal morphology, enzyme activity, and nutrient transporters in weaning piglets. J Anim Sci 96(3):1130-1139. doi: 10.1093/jas/skx003
Round 2
Reviewer 2 Report
Effects of dietary amylose-amylopectin ratio on growth performance and intestinal digestive and absorptive function in weaned piglet response to lipopolysaccharide
This is a second review of the above manuscript. The author took steps and improved their manuscript significantly and provided enough responses to my previous comments. However, the presentations of the work and the English writing of this work still need to improve. I urged authors to have their revised manuscript checked by an English Editing Service to minimize the grammatical, syntax, and word errors. This should be done clearly. This is not the role of the reviewer to state all English wording issues in the text. Also, the authors were careless about formatting their paper based on the journal guideline again in many ways.
Comments
Simple summary
L28: Our result indicated that
L29: dietary amylose/amylopectin ratio of 0.60 can reduce the feed… >>> dietary amylose/amylopectin ratio of 0.60 could reduce feed…
L32: provide data support for the formulation of feed in weaned piglets during bacterial infection>>provide data to support the formulation of feed in weaned piglets during bacterial infection
Abstract
L37: ..their body weight and litters …>>..their body weight (BW) and litters …
L39: day >>>d
L43: body weight>>.BW
Please use BW abbreviations once it defined first
P-value should be small not capital
L54: …can improve…>>>… could improve…
Introduction
L71: “It is well-known fact that weaning is an important affair for the life of pigs”: Please provide at least two references for this statement. Plus: It is well documented that weaning is an important affair for the life of pigs
M&M
L131: Headline and sub-headline initials should be capitalized.
The References are not formatted well based on the journal guideline again. Please double-check the formatting. Abbreviations of journals’ names should be used, the year should be bold etc…
Author Response
Dear Amadeusz Bryła
Thank you very much for your and reviewers’ constructive comments concerning our manuscript entitled “Effects of dietary amylose-amylopectin ratio on growth performance and intestinal digestive and absorptive function in weaned piglet response to lipopolysaccharide” (animals-1723162). Those criticisms are all valuable and very helpful for improving our paper. We have carefully revised our manuscript in accordance with your kind advices and reviewers’ detailed comments. The amendments of reviewers were highlighted in paper. We would like to submit a revision for your consideration again. Looking forward to hearing from you soon.
With best regards,
Sincerely,
Huansheng Yang Ph.D.
e-mail: [email protected]
This is a second review of the above manuscript. The author took steps and improved their manuscript significantly and provided enough responses to my previous comments. However, the presentations of the work and the English writing of this work still need to improve. I urged authors to have their revised manuscript checked by an English Editing Service to minimize the grammatical, syntax, and word errors. This should be done clearly. This is not the role of the reviewer to state all English wording issues in the text. Also, the authors were careless about formatting their paper based on the journal guideline again in many ways.
Response: Thank you very much for your sincere comment. According to your suggestion, we have asked native English-speaking colleagues to modify the grammar of the article.
Comments
Simple summary
L28: Our result indicated that
Response: Thank you very much for your sincere comment. We had revised it in Line 28.
L29: dietary amylose/amylopectin ratio of 0.60 can reduce the feed… >>> dietary amylose/amylopectin ratio of 0.60 could reduce feed…
Response: Thank you very much for your sincere comment. We had revised it in Line 31-32.
L32: provide data support for the formulation of feed in weaned piglets during bacterial infection>>provide data to support the formulation of feed in weaned piglets during bacterial infection
Response: Thank you very much for carefully inspection. We had revised it in Line 37.
Abstract
L37: ..their body weight and litters …>>..their body weight (BW) and litters …
Response: Thank you very much for carefully inspection. We had revised it in Line 37.
L39: day >>>d
Response: Thank you very much for your sincere comment. We had revised it in Line 39.
L43: body weight>>.BW
Response: Thank you for your careful observation. We had revised it in line 43.
Please use BW abbreviations once it defined first
Response: Thank you for your careful suggestion. We had revised them in the line 37 and 125.
P-value should be small not capital
Response: Thank you for your careful suggestion. We had revised them in the text.
L54: …can improve…>>>… could improve…
Response: Thank you very much for your sincere comment. We had revised it in Line 54.
Introduction
L71: “It is well-known fact that weaning is an important affair for the life of pigs”: Please provide at least two references for this statement. Plus: It is well documented that weaning is an important affair for the life of pigs
Response: Thank you very much for your sincere comment. We had revised it in Line 72, 380 and 383.
M&M
L131: Headline and sub-headline initials should be capitalized.
Response: Thank you for your sincere suggestion. We had revised it in Line 132.
The References are not formatted well based on the journal guideline again. Please double-check the formatting. Abbreviations of journals’ names should be used, the year should be bold etc
Response: Thank you very much for your sincere comment. We had revised it in References.